# Endophytes from African Rice (*Oryza glaberrima* L.) Efficiently Colonize Asian Rice (*Oryza sativa* L.) Stimulating the Activity of Its Antioxidant Enzymes and Increasing the Content of Nitrogen, Carbon, and Chlorophyll

**DOI:** 10.3390/microorganisms9081714

**Published:** 2021-08-11

**Authors:** Carmen Bianco, Anna Andreozzi, Silvia Romano, Camilla Fagorzi, Lisa Cangioli, Pilar Prieto, Fousseyni Cisse, Oumar Niangado, Amadou Sidibé, Silvia Pianezze, Matteo Perini, Alessio Mengoni, Roberto Defez

**Affiliations:** 1Institute of Biosciences and BioResources, Via P. Castellino 111, 80131 Naples, Italy; anna.andreozzi@ibbr.cnr.it (A.A.); romanosilvia067@gmail.com (S.R.); roberto.defez@ibbr.cnr.it (R.D.); 2Department of Biology, University of Florence, Via Madonna del Piano 6, 50019 Sesto Fiorentino, Italy; camilla.fagorzi@unifi.it (C.F.); lisa.cangioli@unifi.it (L.C.); alessio.mengoni@unifi.it (A.M.); 3Departamento de Mejora Genética, Campus ‘Alamedadel Obispo’, Instituto de Agricultura Sostenible (IAS), Consejo Superior de Investigaciones Científicas (CSIC), Avd. Menéndez Pidal s/n, 14004 Córdoba, Spain; pilar.prieto@ias.csic.es; 4Institut d’Economie Rurale, Rue Mohamed V Bamako, Bamako B.P. 258, Mali; fousscisse@yahoo.fr (F.C.); amadousidibe57@yahoo.fr (A.S.); 5Syngenta Foundation for Sustainable Agriculture, Bamako B.P.E. 1449, Mali; oumar.niangado@syngenta.com; 6Fondazione Edmund Mach, Via Mach 1, 38098 San Michele All’Adige, Italy; silvia.pianezze@fmach.it (S.P.); matteo.perini@fmach.it (M.P.); 7Environmental and Animal Sciences DI4A, Università degli Studi di Udine, Via Sondrio 2/A, 33100 Udine, Italy

**Keywords:** nitrogen fixation, indole-3-acetic acid (IAA), 1-aminocyclopropane-1-carboxylic acid (ACC) deaminase, rice endophytic bacteria, salt stress, anti-oxidative enzymes

## Abstract

Bacterial endophytes support the adaptation of host plants to harsh environments. In this study, culturable bacterial endophytes were isolated from the African rice *Oryza glaberrima* L., which is well-adapted to grow with poor external inputs in the tropical region of Mali. Among these, six N-fixer strains were used to inoculate *O. glaberrima* RAM133 and the Asian rice *O. sativa* L. cv. Baldo, selected for growth in temperate climates. The colonization efficiency and the N-fixing activity were evaluated and compared for the two rice varieties. *Oryza sativa*-inoculated plants showed a fairly good colonization efficiency and nitrogenase activity. The inoculation of *Oryza sativa* with the strains *Klebsiella pasteurii* BDA134-6 and *Phytobacter diazotrophicus* BDA59-3 led to the highest nitrogenase activity. In addition, the inoculation of ‘Baldo’ plants with the strain *P. diazotrophicus* BDA59-3 led to a significant increase in nitrogen, carbon and chlorophyll content. Finally, ‘Baldo’ plants inoculated with *Kl. pasteurii* BDA134-6 showed the induction of antioxidant enzymes activity and the maintenance of nitrogen-fixation under salt stress as compared to the unstressed controls. As these endophytes efficiently colonize high-yielding crop varieties grown in cold temperate climates, they become good candidates to promote their growth under unfavorable conditions.

## 1. Introduction

Endophytic microbes are an important component of Plant Growth Promoting Bacteria (PGPB), are ubiquitously present in all plants and do not cause disease symptoms [1]. They are often adapted to extreme habitats, developing strategies to induce stress responses in the host plants in which they reside [2]. The endophytes, being able to colonize the internal plant tissues, communicate and interact with the host plant more efficiently than rhizospheric colonizers. In addition, once they have established within the tissues of the host plants, they are no longer subjected to the variation of changing soil conditions [1,2]. The endophytic PGPB are currently attracting a good deal of research interest due to their ability to help their host plants in getting increased amounts of limiting plant nutrients, including nitrogen, iron, and phosphorus, and to protect them against abiotic and biotic stresses, thus promoting their growth [3]. Soil salinity, more than any other factor, imposes a major constraint to the yield of agricultural crops, such as maize, rice, and sugarcane. It affects plant growth by increasing stress factors, such as ethylene, Na^+^, and reactive oxygen species (ROS), which are harmful to the plant physiology, thus leading to growth defects [4,5].

To protect itself from oxidative damage, the plant produces anti-oxidative enzymes, which scavenge excessive ROS generated during salinity stress [6]. These enzymes include superoxide dismutase (SOD), catalase (CAT), peroxidase (POX), and ascorbate peroxidase (APX). Under stress conditions, the enhanced activities of anti-oxidative enzymes have been reported as a tolerance mechanism in plants [7]. Endophytic PGPB can assist plant growth under salinity stress through the stimulation of growth and the action of 1-aminocyclopropane-1-carboxylate (ACC) deaminase. ACC deaminase, encoded by the *acdS* gene, is responsible for the breakdown of ACC, the direct precursor of ethylene in all higher plants, into ammonia and α-ketobutyrate, used by bacteria as nitrogen and carbon sources [8,9]. The increase of ACC deaminase activity by overexpression of the corresponding gene has been reported to relieve environmental stresses of the host plants as compared to plants inoculated with the wild-type strains or the uninoculated ones [10].

Rice, a staple food and primary crop grown worldwide, feeds almost half of the world’s population, particularly in Asia, Africa and Latin America. There are two distinct types of domesticated rice: *O. sativa* L. (Asian rice) and *O. glaberrima* Steud (African rice). African rice was domesticated about 2000–3000 years ago from the wild ancestor *O. barthii* by people living in the floodplains of the upper Inland Delta of the Niger River (Mali) [11]. Morpho-agronomic and molecular characterization of *O. glaberrima* accessions confirmed the existence of a relatively high reservoir of genetic diversity in this area [12]. Although there are farmers in West Africa still cultivating *O. glaberrima*, due to its low performance caused by the lodging and shattering, it is being massively replaced by Asian rice varieties. However, with a minimal intervention of humans and without the addition of nitrogen fertilizers, *O. glaberrima* acquired and developed tolerance and resistance against the majority of the biotic and abiotic stresses affecting rice cultivation in Sub-Saharan Africa [13,14]. Previous studies showed that rice plants (*O. sativa* L. cv. Baldo) adapted to temperate climates and were endophytically colonized by bacterial strains having plant growth-promoting (PGP) traits, which had positive changes in chlorophyll and nitrogen content, and shoot dry weight compared to uninoculated control plants [15], suggesting that auxin production and nitrogen-fixing abilities could be relevant traits of PGPB with a potential application in rice. 

In this study, for the first time, the African rice *O. glaberrima* was selected to isolate cultivable bacterial endophytes. Considering that the root-associated microbes are affected by geographical location, soil source, host genotype, cultivation practice, and environmental stresses [16,17], we suppose that the population of bacterial endophytes hosted by *O. glaberrima* rice cultivated in the Niger River valley should be different from the one hosted by *O. sativa*, a tropical cereal extensively bred in Europe, and subjected to long-term agrochemical applications, and was selected for growth in a temperate climate with cold winter. Therefore, the main objective of this study was to establish a collection of relevant endophytes. To achieve this goal, we planned to: (1) isolate cultivable endophytic bacteria from the tissues of *O. glaberrima* plants (primary host); (2) characterize the selected endophytes for their plant growth-promoting (PGP) traits such as IAA-production, nitrogen-fixation and presence of the *acdS* gene, coding for ACC-deaminase enzyme; (3) analyse the ability of the N-fixing bacteria to colonize *O. sativa* plants (secondary host) and estimate the effect of their inoculation on growth parameters of the host plants; 4) evaluate the response of inoculated plants to salt stress.

## 2. Materials and Methods

### 2.1. Isolation of Endophytic Bacteria from O. glaberrima Rice Tissues

*O. glaberrima* plants were kindly provided by Fousseyni Cisse and Amadou Sidibé from Institut d’Economie Rurale, Bamako. The Inner Niger Delta suffers from degraded soil quality and severe environmental stresses. The most important stress is the variation in the water regime, which is caused by both climate variability and upstream use of water. Endophytic bacteria were isolated from whole, healthy *O. glaberrima* rice plants grown up to 10 days in the Inner Niger Delta (Mali). 

Plants were surface sterilized as follows: 1 min in 70% ethanol, 1 min in 5% sodium hypochlorite (NaClO), 30 s in 70% ethanol, and washing several times with sterile distilled water [15]. When sterilized plants were placed on LB agar plates and incubated at 30 °C for 3 days, no microbial growth was observed around the plant tissues. Sterilized samples were then ground with 5 mL of 1× PBS buffer using a sterile mortar and pestle. For the isolation of the endophytic bacteria, the tissue extracts were serially diluted in 1× PBS solution and plated onto R-2A plates (Sigma-Aldrich, St. Louis, MO, USA). The plates were incubated for up to 5 days at 30 °C to isolate fast-growing endophytes. Single colonies with different sizes, shapes and colors were selected and then streaked onto new R-2A plates. This procedure was repeated several times in order to isolate purified colonies. All purified colonies were grown in R-2A broth (HiMedia, VWR, Radnor, PA, USA) at 30 °C for 2 days. The bacterial cultures in exponential growth phase (OD_600_ ± 0.7) were stored at −80 °C in a solution containing 16% (*v*/*v*) dimethyl sulfoxide (DMSO) and 10% (*v*/*v*) glycerol [18].

### 2.2. Bacterial Identification by 16S rRNA Sequence Analysis

#### 2.2.1. DNA Extraction 

The genomic DNA from each strain’s culture was isolated using the Wizard^®^ Genomic DNA Purification Kit (Promega, Madison, WI, USA), following the manufacturer’s instructions. The quality of DNA samples was analysed by 1% agarose gel electrophoresis. For DNA quantification 2 µL of the extracted genomic DNA were analysed by the NanoDrop 2000 spectrophotometer (Thermo Scientific, Walthman, MA, USA).

#### 2.2.2. PCR Amplification of the 16rRNA Gene Fragment 

To amplify the 16S rRNA gene fragment (1.4 Kb), universal primer pairs were used: 8-27F primer (5′-AGA GTT TGA TCC TGG CTC AG-3′), and 1510-1492R primer (5′-ACG GCT ACC TTG TTA CGA CTT-3′). The amplification reaction (25 µL) contained: 2.5 μL of 10x DreamTaq Buffer (Thermo Scientific, Walthman, MA, USA), 2 mM MgCl_2_, 100 ng of the genomic DNA, 200 nM of each primer, 1 mM of each dNTP, and 0.03 U of DreamTaq Hot Start DNA Polymerase (Thermo Scientific, Walthman, MA, USA). A PCR mix without DNA was included as a negative control. The PCR reaction conditions were as follows: 3 min at 95 °C, 30 cycles of denaturation at 95 °C for 30 s and annealing at 52 °C for 30 s, extension at 72 °C for 1 min, and a final extension at 72 °C for 10 min [18].

#### 2.2.3. Cloning and Sequencing of the 16S Fragment

The amplified 16S rRNA gene fragments were introduced into a linearized pHTP1 vector and cloned into competent cells of *E. coli* DH5α using the NZYEasy Cloning and Expression Kit I (NZYTech, Lisboa, Portugal). The 16S rRNA gene inserts were prepared by PCR using primers with 16bp overhangs included on 5′-ends of both forward and reverse primers, in order to provide vector-complementary single-strands terminals. The following primers were designed: 16S-NZYf (5′-T CAG CAA GGG CTG AGG CCT AGA GTT TGA TCA TGG CTC AG-3′) and 16S-NZYr (5′-T CAG CGG AAG CTG AGG GGT TAC CTT GTT ACG ACT T-3′). The preparation of DNA inserts was carried out with a PCR amplification reaction (25 μL) containing 0.5 µL of 16S rRNA gene PCR products, above obtained, 200 nM of the 16S-NZY primer pairs, 2.5 μL of 10x Buffer, 2 mM MgCl_2_, 1 mM of each dNTP, 0.03 U of DreamTaq Hot Start DNA Polymerase (Thermo Scientific, Walthman, MA, USA). The PCR reaction conditions were as follows: 3 min at 95 °C, 30 cycles of denaturation at 95 °C for 30 s and annealing at 68 °C for 30 s, extension at 72 °C for 1 min, and a final extension at 72 °C for 10 min. The inserts were purified using the QIAquick PCR Purification Kit (QIAGEN, Hilden, Germany) and transferred into pHTP1 vector.

A ligase-independent cloning reaction was prepared following the manufacturer’s instruction, using a vector to insert molar ratio of 1 to 5. The cloning product was used to transform competent cells of *E. coli* DH5α. The recombinant clones were identified by colony PCR reactions. For the amplification of the insert, the 16S-NZYf and 16S-NZYr primers pair and the PCR reaction conditions described above were used. The PCR products were purified, and their nucleotide sequences determined through DNA fragments sequencing by fluorescent chain termination and capillary electrophoresis method (Eurofins, Ghent, Belgium). The obtained sequences were first trimmed to eliminate the low-quality ends and then compared with those deposited in GenBank database (www.ncbi.nlm.nih.gov/BLAST/, accessed on 25 March 2020) using the BLASTN algorithm (BLAST Local Alignment Search Tolls).

#### 2.2.4. Phylogenetic Characterization

The 16S rRNA gene sequences were first aligned by using ClustalW and then analysed by MEGA6 software [19] to infer the molecular phylogeny by the maximum likelihood method based on Tamura–Nei model [20]. The robustness of the phylogenetic tree was evaluated by bootstrap analysis of 1000 pseudo-replications. The scale bar on the rooted tree indicates 0.05 substitutions per nucleotide position.

### 2.3. Genome Sequencing and Bioinformatic Analysis

DNA was extracted from overnight grown cultures in LB medium by using PowerSoil DNA Isolation Kit (Qiagen, Hilden, Germany) After gel electrophoresis and fluorometric quantification (Qubit, Thermo Fisher Scientific, Waltham, MA, USA), genomic DNA was fragmented with g-TUBE (Covaris Inc., Woburn, MA, USA) to average 15 Kbp size and used for library preparation using Pacific Biosciences SMRTbell Express Template Prep Kit 2.0 (Pacific Biosciences, Menlo Park, CA, USA). Samples were sequenced with the Sequel Sequencing Kit 3.0 in a Sequel apparatus (Pacific Biosciences, Menlo Park, CA, USA) by SMRT technology [21]. Obtained reads were analysed using SMRT Link software ver. 8.0.0.80529 (Pacific Biosciences, Menlo Park, CA, USA) producing *oriC*-oriented assembly by running the “microbial multiplexing” pipeline with default options. Annotation and the COG assignment were performed by Prokka 1.4.0 [22] from the Galaxy webserver Orione (https://orione.crs4.it/, accessed on 1 April 2021) [23]. Taxonomic identification of strains from genome sequence was performed using ContEst16S [24] and Type (Strain) Genome Server [25]. Presence of CRISPR arrays was checked by CRISPRFinder [26]. Secondary metabolites gene clusters have been identified by AntiSMASH [27]. The presence of gene modules for chemotaxis, flagellar biosynthesis and secretion systems were inspected by performing KAAS functional annotation [28]. The presence of Type III secretion systems was additionally anayzed by T3Sepp [29]. Sequences are deposited in NCBI database under Bioproject (PRJNA670042).

### 2.4. Screening of Plant Growth-Promoting (PGP) Activities

#### 2.4.1. Indole-3-Acetic Acid (IAA) Production

Salkowski colorimetric assay. The IAA production into bacterial cultures was analysed by using the Salkowski reagent [30] as described below. Bacterial culture was grown overnight in LB medium containing 100 μM L-tryptophan at 30 °C. Liquid culture was centrifuged at 12,000 rpm for 5 min to collect the bacterial cells. The culture supernatant was then mixed with the Salkowski reagent (35% HClO_4_ 49 mL and 0.5 M FeCl_3_ 1 mL) in a 1 to 1 ratio and incubated in the dark at room temperature. After 30 min of incubation, the absorbance of the mixtures was estimated at 530 nm. The bacterial cells were dried and weighed for data normalization. The concentration of IAA from each culture medium was calculated from a pure IAA standard curve. Data are the mean ±SD of at least five independent bacterial cultures each grown at different times and from different bacterial colonies.

#### 2.4.2. HPLC—Tandem Mass Spectrometry (HPLC-MS/MS)

The high-pressure liquid chromatography (HPLC) column was an Ultra AQ C18 (3 µm pore size, 100 × 2.1 mm), operating at 250 µL/min flow rate and thermostated at 35 °C. The mobile phases were 5 mM ammonium acetate in LC-MS grade H_2_O containing 0.2% formic acid (mobile phase A) and acetonitrile containing 0.2% formic acid (mobile phase B). For gradient elution, the percentage of mobile phase B was linearly increased from 0% to 40% in 11 min, then to 90% in 2 min; after 7 min at 90% mobile phase B, the initial composition was restored and the system equilibrated for 10 min. The injection volume was 10 µL.

The HPLC was a Series 200 pump with a Series 200 autosampler (Perkin Elmer, Milano, Italy) coupled to a 4000 Q Trap mass spectrometer (Sciex, Old Connecticut Path Framingham, MA, USA), equipped with a Turbo V IonSpray interface. The interface and MS parameters were optimized by infusion of a standard solution of IAA (Sigma-Aldrich, St. Louis, MO, USA). The settings used were the following: curtain gas 25, nebulizer gas 70, heater gas 68, temperature 550 °C, ESI spray voltage, 5.2 kV; declustering potential (DP) 40 V, entrance potential (EP) 10 V. Nitrogen was used as curtain, nebulizer, and heater gas. Acquisition was done in multiple reaction monitoring (MRM) mode in positive ion polarity, recording two MRM transitions for IAA and its isotopologue, indole-2,4,5,6,7-d5-3-acetic-2,2-d2 acid (d7-IAA; purchased from CDN Isotopes, Pointe-Claire, Canada): the protonated ions of IAA *(m/z* 176.1) and d7-IAA (*m/z* 183.1) were fragmented in the collision cell of the MS and the product ions at *m/z* 130 (collision energy, CE, 23 V) and *m/z* 103 (CE 36 V) for IAA, and *m/z* 136 (CE 23 V) and *m/z* 109 (CE 40 V) for d7-IAA were recorded. For HPLC-MS/MS analyses, 1 µL of a solution of d7-IAA (25 ng/µL in methanol) was added to 100 µL of sample and then injected in the HPLC-MS/MS instrument. Peak identification was accepted only if the relative retention time of IAA was consistent with that of the internal standard d7-IAA (±0.5%) and if the ratio of the signals of the two transitions were in agreement with that recorded for the standard. For quantitative purposes, ratio of the peak areas (PAR, peak area ratio) of the signals corresponding to the most intense MRM transition for IAA (*m/z* 176 → *m/z* 130) and for d7-IAA (*m/z* 183 → *m/z* 136) was measured for each sample. The sample concentration was calculated using a six-point linear calibration curve prepared adding scalar amounts of IAA to several aliquots of blank culture medium to mimic actual samples in the concentration range from 0.025 ng/mL to 0.5 ng/mL. The calibration points were processed in parallel to the samples, following the same procedure. For each sample three 100 µL aliquots were measured in triplicate.

### 2.5. PCR Amplification of the Nitrogenase Iron Protein (nifH) and ACC Deaminase (acdS) Genes Fragments

The PCR amplification of the *nifH* and *acdS* genes was performed to identify nitrogen-fixing (diazotrophic) and ACC-deaminase-producing (halotolerant) bacteria. The primer pairs used were: 19F (5′-GCI WTY TAY GGI AAR GGI GG-3′) and 407R (5′-AAI CCR CCR CAI ACI ACR TC-3), which amplified a 388 bp fragment of the *nifH* gene [31] acdSF3 (5′- ATCGGCGGCATCCAGWSNAAYCANAC -3′) and acdSR3 (5′-GTGCATCGACTTGCCCTCRTANACNGGRT-3), which amplified a segment of 683 bp of the *acdS* gene [32]. The amplification reactions (25 µL) for the screening of the *nifH* and *acdS* genes contained: 2.5 μL of 10x DreamTaq Buffer (Thermo Scientific, Walthman, MA, USA), 2 mM MgCl_2_, 100 ng of the genomic DNA, 200 nM (*nifH)* and 400 nM (*acdS*) of each primer, 1 mM of each dNTP, and 0.03 U of DreamTaq Hot Start DNA Polymerase (Thermo Scientific, Walthman, MA, USA). The thermocycling conditions for *nifH* and *acdS* genes were as follows: 30 sec at 94 °C, 35 cycles of denaturation at 95 °C followed by annealing at 50 °C (*nifH*) and 62 °C (*acdS*) for 1 min, extension at 72 °C for 1 min, and a final extension at 72 °C for 10 min. Sterile Milli-Q water, instead of genomic DNA, was used to set up control reaction. As previously proposed by Zhengyi et al. [32], the DNA of *Herbaspirillum seropedicae* z67 (HSz67) was used as positive control for *acdS* gene fragments, while the DNA of *Sinorhizobium meliloti* 1021 was used as positive control for *nifH* gene fragments. The DNA of the *Escherichia coli* MG1655 was used as negative control for both genes.

### 2.6. Plant Growth and Inoculation Methods

‘Baldo’ rice has been selected to growth in Northern Italy (Piemonte and Lombardia regions) characterized by environmental conditions very different from the ones typical of West Africa. Indeed, the average annual temperature in Mali is around 27 °C, with few variations in different seasons, while in the regions of northern Italy temperatures fluctuate between 0 °C in winter and an average of 20 °C in summer. *O. sativa* L. cv. Baldo seeds were kindly provided by Stefano Monaco and Elisa Zampieri from CREA-CI, Research Centre for Cereal and Industrial Crops, Vercelli, Italy. Dehulled seeds of ‘Baldo’ and *O. glaberrima* RAM133 (Mali, West Africa) were surface sterilized as described by Defez et al. [18] with the following changes: 70% ethanol for 7 min and 5% sodium hypochlorite solution containing 0.1% tween 20. Seeds were then washed several times with sterilized distilled water, positioned onto the surface of 0.8% water-agar plates and incubated at 21 °C in the dark for germination [18].

After 5 days, the roots of the germinated seeds were cut (0.5 cm from the bottom) with a sterile bistoury and incubated in Petri dishes with 50 mL of 1× PBS solution containing each strain to a final concentration of 10^6^ cells mL^−1^ for 4 h at room temperature [15]. Seeds incubated in 1× PBS were used as control. Infected seeds were then transferred into plastic pots containing sand (2–1.2 mm granule size) and perlite (3–4 mm granule size) soil in 1:1 ratio, or into hydroponic units (plastic tubes of 15 cm in length and 5 cm in diameter) containing Kimura B nitrogen-free medium reported in [33] with some modifications.

The macronutrients were the following: KH_2_PO_4_ (0.2 mM), KNO_3_ (0.01 mM), K_2_SO_4_ (0.1 mM), CaSO_4_ (0.4 mM), MgSO_4_·7H_2_O (0.5 mM). The micronutrients were the following: Fe-EDTA (0.11 mM), MnSO_4_ (1.8 μM), H_3_BO_3_ (46 μM), ZnSO_4_·7H_2_O (0.3 μM), and CuSO_4_·5H_2_O (0.3 μM). Each planting unit was kept in the growth chamber under long daylight (16 h), 19–23 °C temperature and 75% relative humidity.

### 2.7. Re-Isolation of Diazotrophic Bacteria from Inoculated Plants

The colonization of *O. sativa* cv. Baldo and *O. glaberrima* RAM133 plants by the selected endophytes was assessed at 14 days after inoculation (DAI) for plants grown in sand-perlite soil. The surface of the whole plants was sterilized using the following procedure: (i) 1 min in 70% EtOH; (ii) washing several times with sterile distilled water; (iii) incubation for 5 min in 5% sodium hypochlorite solution containing 0.1% tween 20; (iv) washing several times with sterile distilled water. The tissues of the whole plant were then homogenized in 5 mL of 1× PBS using a sterile mortar and pestle. Dilutions of the homogenates (1:100 for the extract of Baldo and 1:1000 for RAM133) were spread onto 1.5% LB agar plates containing the specific antibiotics (60 µg mL^−1^ ampicillin for *P. diazotrophicus* BDA59-3; 25 µg mL^−1^ carbenicillin for *Ko. pseudosacchari* BDA62-3, *Kl. pasteurii* BDA134-6 and *Ko. oryzendophytica* BDA137-1; 25 µg mL^−1^ penicillin G for *E. sacchari* BDA86-11; 50 µg mL^−1^ novobiocin for *Enterobacter* sp. BDAM41-2) and incubated at 30 °C. Specific antibiotics were selected after analysing the resistance of the selected strains to different antibiotics (Appendix A). After 24 h, the CFU number was counted. The identity of re-isolated endophytes was verified by their antibiogram patterns. To confirm that the sterilization process described above was successful, sterilized plant tissues were placed on 1.5% LB agar plates and incubated at 30 °C for 3 days. After the incubation period, no microbial growth was observed around the plant tissues. Data for plants grown in sand-perlite soil and hydroponic conditions are the mean ± SD of at least five and eight independent replicates, respectively, each carried out at different times.

### 2.8. Strains Tagging with FPs

Plasmids pMP4655 (harbouring the EGFP marker) [34,35] were purified from *Escherichia coli* DH5α host cells using the PureLink™ HiPure Plasmid Filter Midiprep Kit (Thermo Scientific, Walthman, MA, USA) according to the manufacturer’s instructions. The transformation of the strains BDA62-3 and BDA134-6 was carried out by electroporation method, while that of the strain BDA59-3 was done by triparental mating. Protocols for the electrocompetent cells preparation, the electroporation and the triparental mating were those reported in [15].

### 2.9. Confocal Laser Scanning Microscopy Detection

To elucidate the colonization process of in vitro rice plants by BDA62-3, BDA134-6 and BDA59-3 strains, one plant per bacteria treatment was analysed each day from 15 days after bacterization. At least ten different whole roots from each rice plant were mounted with distilled water for confocal visualization. Confocal laser scanning microscopy (CLSM) stacks were obtained using an Axioskop 2 MOT microscope (Carl Zeiss Inc., Göschwitzer Straße, Jena GmbH, Germany) equipped with a krypton and an argon laser and supported by the Laser Scanning System LSM5 PASCAL software (Carl Zeiss Inc., Göschwitzer Straße, Jena GmbH, Germany). For microscope data analysis, the Zeiss LSM Image Browser version 4.0 program (Carl Zeiss Inc., Göschwitzer Straße, Jena GmbH, Germany) was used. Bacterial colonization of rice roots and shoots was analyzed from 3-D confocal data stacks. Projections from adjacent confocal optical sections were made for composing images shown in this study. Final figures were processed with Photoshop 4.0 software (Adobe Systems Inc., San Jose, CA, USA).

### 2.10. Nitrogenase Activity by Acetylene Reduction Assay (ARA)

#### 2.10.1. Bacterial Cultures

Strains selected as positives in the PCR analysis of *nifH* gene were grown in 1x M9 minimal ammonium chloride medium [36] for 48 h at 30 °C on a rotary shaker (200 rpm). The cell pellets (2 mL) were suspended in the same medium without nitrogen source, as described by Defez et al. [36] and transferred into glass tubes airtight with a serum cap. To assess nitrogenase activity, we used the acetylene reduction assay (ARA) [37]. Purified acetylene was injected into each tube at approximately 10% of headspace volume under a hypoxic atmosphere (2% O2). After 18 h of incubation at 30 °C, a 1 mL sample from each bottle was injected into a gas chromatograph GC Clarus^®^580 (Perkin-Elmer, Walthan, MA, USA) to evaluate the amount of the produced ethylene (C_2_H_4_) after the addition of acetylene. A parallel set of C_2_H_4_ standards was injected into the GC to calibrate its sensitivity, linearity, and a concentration of C_2_H_4_ produced in samples. The GC used was equipped with a hydrogen flame detector (FID) and a TG-IBOND Alumina (Na_2_SO_4_ deactivate) column (30 m × 0.53 mm × 10 μm) (Thermo Scientific, Walthman, MA, USA). The flow rate of the carrier (helium) was 48 cm s^−1^, and the oven programmer was 130 °C isocratic for 3 min. Inoculated tubes without injected acetylene were used as a negative control. Data are expressed as nmol ethylene mg cells^−1^ min^−1^ and are the mean ± SD of at least five independent bacterial cultures, each grown at different times and from different bacterial colonies.

#### 2.10.2. Inoculated Plants

Two-week-old rice plants (*O. sativa* cv. Baldo and *O. glaberrima* RAM133), grown as previously described, were carefully removed from the pots and the roots rinsed with water. The plants were then transferred into 20 mL glass tubes, incubated in a greenhouse for 20 h and analysed for the ARA assay as described by Defez et al. [18]. The amount of ethylene produced was measured by gas chromatography as described above. Data for plants grown in sand-perlite soil and hydroponic conditions are expressed as nmol ethylene plant^−1^ min^−1^ and are the mean ± SD of at least six and eight independent replicates, respectively, each carried out at different times.

### 2.11. ^15^N_2_ Incorporation

Two-week-old rice plants *(O. sativa* cv. Baldo) non-inoculated and inoculated with the strains BDA59-3, BDA62-3 and BDA134-6 were removed from the pots and washed several times with sterilized distilled water. Non-inoculated plants were used as references. 

The plants were placed into 20-mL bottles containing 2 mL of minimal medium free of nitrogen sources and sealed with a rubber serum stopper. The bottles containing an atmosphere of 15% (vol/vol) ^15^N_2_ (99.9 atom%), 20% (vol/vol) N_2_, and 10% (vol/vol) O_2_ were incubated in greenhouse for 24 h in the dark. The plants were then taken from the bottles, freeze-dried and ground in a mortar. The ^15^N/^14^N ratios were measured using an isotope mass spectrometer (Elementar Gmbh, Isoprime Ltd., Langenselbold, Germany) after total combustion in an elemental analyser (VARIO CUBE, Isoprime Limited, Langenselbold, Germany). To analyse the samples the amount weighed was 3.5 mg. The samples were analysed in triplicate and reported as an average. 

Due to the small variations recorded, the data is not reported as atom% in excess with respect to the non-inoculated samples but, according to the IUPAC protocol, they are denoted in delta in relation to the international standard Air (atmospheric N_2_ with 0.366 atom%) for δ^15^N, according to the following general equation: δi E = (i RSA–i RREF)/ i RREF [38], where *i* is the mass number of the heavier isotope of element E (^15^N); *RSA* is the respective isotope ratio of a sample (such as number of ^15^N atoms/number of ^14^N atoms or as approximation ^15^N/^14^N); *RREF* is the respective isotope ratio of internationally recognized reference material. The delta values are multiplied by 1000 and expressed in units “per mil” (‰). The isotopic values were calculated against working in-house standards, which were themselves calibrated against international reference materials: L-glutamic acid USGS 40 with δ^15^N = −4.52‰ (U.S. Geological Survey, Reston, VA, USA) and potassium nitrate IAEA-NO3 (δ^15^N = +4.7‰) from IAEA (International Atomic Energy Agency, Vienna, Austria). The uncertainty of measurements, calculated as standard reproducibility obtained from the analysis of the same sample over time and multiplied for the coverage factor 2, was <0.3‰ for δ^15^N.

### 2.12. Salt Stress

To apply salt stress, three-week-old plants were carefully removed from the pots and the roots rinsed with water. Plants were then transferred into 20 mL glass tubes airtight with serum cap containing 2 mL of minimal medium supplemented with NaCl at a final concentration of 0.3 M and free of nitrogen sources. Plants were then incubated in greenhouse for 20 h and analysed for the ARA test as described by Defez et al. [18]. After the ARA test, plant tissues (roots and leaves) were immediately collected, frozen in liquid nitrogen and stored at −80 °C for further analyses.

### 2.13. Antioxidant Enzymes Assays

The soluble leaf proteins were extracted and quantified as described in Defez et al. [18]. The activities of all enzymes [ascorbate peroxidase (APX), catalase (CAT), peroxidase (POX) and superoxide dismutase (SOD)] were measured spectrophotometrically at 25 °C within the linear region for both time and enzyme concentration as reported in Bianco and Defez [39].

### 2.14. Chlorophyll Content Measurement

At 14 and 40 DAI detached leaves were incubated at room temperature in a 2 mL tube containing 1.5 mL 80% acetone solution for at least 24 h and then clarified by centrifugation for 5 min at 15,000× *g*. The absorbance of the supernatant was measured at wavelengths 645, 646, and 663 nm (A_645_, A_646_, A_663_) [40] with a DU 800 UV/Visible spectrophotomer (Beckman Coulter, Brea, CA, US). Chlorophyll concentration was estimated following the Arnon’s equations [41] as follows:

Chlorophyll a (μg/mL) = 12.7 (*A*_663_) − 2.69 (*A*_645_)

Chlorophyll b (μg/mL) = 22.9 (*A*_645_) − 4.68 (*A*_663_)

Total chlorophyll (μg/mL) = 20.2 (*A*_645_) + 8.02 (*A*_663_)

Data are the means ± SD of at least five independent biological replicates each carried out at different times.

### 2.15. Total Carbon and Nitrogen Analysis

Total organic carbon and nitrogen content from *O. sativa* leaves at 14 and 40 DAI were determined with dried material by using an elemental analyser (CKIC 5E-CHN 2200, Emme3, Lainate, Milano, Italy) at 950 °C, according to the European standard method UNI EN 15407:2011. Data are the means ±SD of at least five independent biological replicates each carried out at different times.

### 2.16. Statistical Analysis

Five biological replicates were carried out for the measurement of IAA production, endophytes re-isolation from inoculated plants, nitrogenase activity for bacterial cultures, nitrogen content, carbon content, and chlorophyll content. Six biological replicates have been carried out for the activity of nitrogenase and anti-oxidative enzymes in salt-stressed inoculated plants. The evaluation of nitrogenase activity in inoculated plants under hydroponic conditions was performed on eight biological replicates. Statistical analysis of nitrogenase activity, physiological parameters (N, C, and Chl) and anti-oxidative enzymes activity was performed by applying the one-way analysis of variance (ANOVA) and Tukey’s HSD post-hoc test. All data are approximately normally distributed. The VassarStats ANOVA program available at http://vassarstats.net/index.html (accessed on 8 June 2020) was used. The remaining data were subjected to Student’s *t*-test. All results were considered statistically significant when *p* ≤ 0.05.

## 3. Results

### 3.1. Identification of Endophytic Bacteria

We here report the isolation of 69 culturable endophytic bacteria from internal tissues of African rice plants (*O. glaberrima* L.). All bacterial isolates were identified by comparative sequence analysis of 16S rRNA gene partial sequences. For all isolates, a 99–100% similarity with already identified species in the GenBank database was found (Appendix A). The isolates belonged to four phyla: Proteobacteria (79%, Aeromonadales, Burkholderiales, Enterobacteriales, Pseudomonadales and Xantomonadales orders), Firmicutes (7%, Bacillales), Actinobacteria (10%, Actinomycetales), and Bacteroides (4%, Sphingobacteriales and Flavobacteriales) (Figure 1 and Appendix A).

The evolutionary history was inferred by using the maximum likelihood method based on the Tamura–Nei model with 1000 bootstrap replications. The percentage of trees in which the associated taxa clustered together is shown next to the branches. The tree is drawn to scale, with branch lengths measured in the number of substitutions per site. The analysis involved 69 nucleotide sequences. Evolutionary analyses were conducted in MEGA6.

### 3.2. PGP-Traits of Bacterial Endophytes

All isolates were examined for the IAA production and for the presence of genes involved in N-fixation (*nifH*) and in the reduction of ethylene level inside the plant (*acdS*). The results obtained in these analyses are reported in Appendix A. When the L-tryptophan was added to the growth medium the 67% of the bacterial strains produced IAA. The IAA produced by the three N-fixing strains *Kl. pasteurii* BDA134-6, *Ko. pseudosacchari* BDA62-3and *P. diazotrophicus* BDA59-3 was quantified by using liquid chromatography coupled to high-tandem mass spectrometry. High signal-to-noise ratios and, therefore, enhanced sensitivity were obtained by applying this procedure. The deuterated IAA (d7-IAA) molecule was used as an internal standard for quantitative purposes. Appreciable IAA level was measured only for the strain *Kl. pasteurii* BDA134-6 (0.44 ± 0.04 ng mL^−1^). Forthe strains *Ko. pseudosacchari* BDA62-3 and *P. diazotrophicus* BDA59-3 IAA level (<0.025 ng mL^−1^) below the threshold of the instrument was observed. For the strain *P. diazotrophicus* BDA59-3 higher value was detected with the spectrofotometric method as compared to mass spectrometry. This discrepancy could result from the well-known ability of the genus *Citrobacter* (some of them are now classified as *P. diazotrophicus*) [42,43] to produce IAA [44] and from the reaction of the total indole compounds with the Salkowski reagent, which alters the IAA evaluation.

The 388 bp amplicon expected for the *nifH* gene was observed for six (9%) bacterial strains. Positive isolates were *Ko. oryzendophytica* BDA137-1, *Enterobacter* sp. BDAM41-2, *Kl. pasteurii* BDA134-6, *Enterobacter sacchari* BDA86-11, *Ko. pseudosacchari* BDA62-3, and *P. diazotrophicus* BDA59-3The diazotrophic nature of the six *nifH*-positive strains was confirmed by measuring the activity of the nitrogenase enzyme in the bacterial cultures (ARA test). The data reported in Table 1 showed that the strain *Kl. pasteurii* BDA134-6 was the best N-fixer. The strains *E. sacchari* BDA86-11, *Ko. pseudosacchari* BDA62-3 and *P. diazotrophicus* BDA59-3 showed lower activity than the one recorded for *Kl. pasteurii* BDA134-6. The lowest nitrogenase activity was measured for the strains *Ko. oryzendophytica* BDA137-1 and *Enterobacter* sp. BDAM41-2. Concerning the *acdS* gene, amplicons of the expected size (683 bp from *H. seropedicae* z67) were obtained for six isolates belonging to the genera *Enterobacter* (*Enterobacter* sp. BDAM41-2 and *E. roggenkampii* BDA86-6), *Stenotrophomonas* (*S. rhizophila* BDAM41-6), *Ralstonia* (*R. mannitolilytica* BDA137-9), and *Microbacterium* (*M. laevaniformans* BDA137-13 and *Microbacterium* sp. BDA137-16).

### 3.3. Genome Sequences of Kl. pasteurii BDA134-6, Ko. pseudosacchari BDA62-3 and P. diazotrophicus BDA59-3 Confirm the Presence of the Full Nitrogenase Genes Set

Single Molecule Real-Time (SMRT) genome sequencing of strains BDA62-3, BDA134-6 and BDA59-3 strains yielded complete genome sequences (Appendix A). Genome-based taxonomic assignment indicated that BDA134-6 and BDA62-3 belong to the species *Kl. pasteurii* and *Ko. pseudosacchari*, confirming 16SrRNA gene sequences, while BDA59-3 belongs to the species *P. diazotrophicus* (Appendix A), although the 16S rRNA gene sequences showed the best match with members of genus *Citrobacter* (data not shown). *Ko. pseudosacchari* BDA62-3 and *Kl. pasteurii* BDA134-6 harbour two replicons for a total genome size of ~5.0 Mbp and ~6.0 Mbp, respectively, while *P. diazotrophicus* BDA59-3 genome consists of a single chromosome ~5.3 Mbp in size (Figure 2). *Ko. pseudosacchari* BDA62-3 contains a chromosome ~4.9 Mbp in size and a plasmid, named pBDA62-3, of 93,051 bp. *Kl. pasteurii* BDA134-6 has a chromosome of ~5.9 Mbp and a plasmid, named pBDA134-6 of 149,270 bp. *P. diazotrophicus* BDA59-3 shows three CRISPR arrays, containing CRISPR-associated proteins similar to those found in *C. rodentium* ICC168 (74% sequence identity). *Kl. pasteurii* BDA134-6 harboured one CRISPR array on the chromosome; however, no *cas* genes were found. No CRISPR were found in *Ko. pseudosacchari* BDA62-3 genome. The methylated DNA motifs detected by SMRT sequencing technology were mostly due to m6A modification (6-methyl adenine) and concerned with the classical GATC motif [45] and 2 and 4 additional motifs in *Ko. pseudosacchari* BDA62-3 and *P. diazotrophicus* BDA59-3, respectively, some of them compatible with the presence of restriction-modification systems (e.g., CTCGAG corresponds to the *XhoI* recognition site, CTGAAG to *Eco57I*). For *Kl. pasteurii* BDA134-6 a m4C (4-methyl cytosine) modification motif was found (VSAGCTSS) present in more than 60% of sites (corresponding to the *AluI* recognition site AGCT). Genome annotation files are reported as Appendix A. 

When looking for genes involved in nitrogen-fixation ability, shared within the three strains, the full set of genes involved in nitrogenase functioning was found, with 17, 12 and 18 orthologs assigned to nitrogenase complex functioning in *Ko. pseudosacchari* BDA62-3, *Kl. pasteurii* DA134-6, and *P. diazotrophicus* BDA59-3, respectively (Table 2 and Appendix A). In order to evaluate the presence of genetic determinants involved in the ability to associate with plants [46], we screened genome sequences of *Ko. pseudosacchari* BDA62-3, *Kl. pasteurii* BDA134-6 and *P. diazotrophicus* BDA59-3 for genes involved in chemotaxis, flagellum biosynthesis, Type III secretion systems, Type VI secretion system (T6SS), COG1609 and COG0667, which have been found enriched in genomes of plant-associated bacteria [46]. A type-III secretion system was retrieved for *Ko. pseudosacchari* BDA62-3 only. However, no effector proteins were identified. No T6SSs were found in *Ko. pseudosacchari* BDA62-3 and *P. diazotrophicus* BDA59-3, while in *Kl. pasteurii* BDA134-6 orthologs encoding for proteins of Hcp, VgrG, IcmF and ClpV were found, suggesting the presence of a complete set for T6SS in this strain. Both *Ko. pseudosacchari* BDA62-3 and *P. diazotrophicus* BDA59-3 displayed the presence of genes for chemotaxis and flagellum biosynthesis (Dataset S1), which include KO annotation lists and AntiSMASH results and Appendix A. *Kl. pasteurii* BDA134-6 only showed *motB* (K02557) and *fliY* (K02424) genes, deserving further attention on chemotaxis and motility phenotypes. In *Ko. pseudosacchari* BDA62-3, *Kl. pasteurii* BDA134-6, and *P. diazotrophicus* BDA59-3 genes assigned to COG1609 were retrieved for 19, 28 and 21 genes, respectively, while for genes assigned to COG0677, only one gene, encoding an UDP-*N*-acetyl-d-mannosamine dehydrogenase, was found in all three genomes (Appendix A). We also looked at the presence of gene clusters involved in the biosynthesis of secondary metabolites potentially involved in plant-microbe and microbe-microbe interactions (Folder S2). A bacteriocin gene cluster (located between 1,505,693–1,516,316 nt) and a nonribosomal peptide-synthetase (NRPS) gene cluster (located between 95,649–147,789 nt), showing 30% similarity with the gene cluster for the biosynthesis of the siderophore turnerbactin [47], was found in *P. diazotrophicus* BDA59-3. In *Ko. pseudosacchari* BDA62-3, a bacteriocin gene cluster (located between 18,651–29,274 nt) was found too, as well as two nonribosomal peptide-synthetase (NRPS) gene clusters (located between 1,305,532–1,359,224 nt and 4,109,397–4,157,770 nt.), showing 30% and 40% similarity with gene clusters for turnerbactin [47] and the lipopeptide ralsolamycin, respectively. Ralsolamycin has been firstly isolated from *Ralstonia solanacearum* and has been shown to facilitate bacterial entry and induce chlamydospore formation in fungi [48]. In *Kl. pasteurii* BDA134-6, a cluster with 100% similarity with that of the biosynthesis of kleboxymycin [49] (3,081,276–3,127,465 nt), and an additional cluster with 30% similarity with the gene cluster for turnerbactin was found (1,594,136–1,645,420 nt). Very recently, a report on the diffusion of kleboxymycin in different *K. oxytoca* strains was posted [50]. The presence of such a gene cluster in a plant-associated bacterium deserves further attention on a role in endophytic colonization and on gene flow of enterotoxins between environmental and clinical strains of *Klebsiella*.

### 3.4. N-Fixing Endophytes Differently Colonize the Primary (O. glaberrima) and Secondary (O. sativa) Host

The ability of the N-fixing endophytes to individually colonize *O. glaberrima* RAM133 and *O. sativa* cv. Baldo plants have been evaluated by measuring the colony-forming units (CFU) of homogenates from whole inoculated plants. The data reported in Table 3 showed that the N-fixing strains colonize more efficiently *O. glaberrima* plants than the *O. sativa* ones. Different colonization ability of *O. sativa* plants was observed for the selected diazotrophic strains: the highest CFU number was obtained for the strains *Kl. pasteurii* 134-6, *E. sacchari* BDA86-11 and *Ko. pseudosacchari* BDA62-3, while the one estimated for the strain, *P. diazotrophicus* BDA59-3 was about a third lower. The lowest colonization ability was observed for the strains *Ko. oryzendophytica* BDA137-1 and *Enterobacter* sp. BDAM41-2. The strains *Kl. pasteurii* BDA134-6, *Ko. pseudosacchari* BDA62-3, and *P. diazotrophicus* BDA59-3 were selected for further analysis.

### 3.5. Colonization of Rice Roots by the Strains Ko. pseudosacchari BDA62-3, Kl. pasteurii BDA134-6 and P. diazotrophicus BDA59-3

The use of fluorescently tagged bacteria and CLSM enabled to demonstrate that the strains *Ko. pseudosacchari BDA62-3, Kl. pasteurii BDA134-6* and *P. diazotrophicus*
*BDA59-3* effectively colonize roots of in vitro rice plants under gnotobiotic conditions. The entire colonization process by tagged *Ko. pseudosacchari BDA62-3, Kl. pasteurii BDA134-6* and *P. diazotrophicus*
*BDA59-3* derivatives were recorded, from bacteria adhesion to the root surface to the colonization of the intercellular spaces within the cortex of the differentiation zone (Figure 3). The rice root rhizoplane was effectively colonized by the three strains. The bacteria were predominantly detected on the rice root differentiation. 

The observations showed that the cells of all bacteria strains were attached along the whole root surface, mostly at random distribution, displaying no preferred site for the external colonization (Figure 3a–c). Internal colonization of root hairs was also detected (Figure 3a–c). Inner colonization of root hair cells did not occur frequently and only a low proportion was internally colonized. From epidermal cells, bacteria were always detected in the intercellular spaces of the root cortex and vascular tissues, never inside cortical cells or invading the interior of the vascular tissue for the three analysed strains (Figure 3d–f). No significant changes were observed during the bioassay for all the rice plants analysed. It is worth mentioning that the tagged bacteria *Ko. pseudosacchari BDA62-3, Kl. pasteurii BDA134-6* and *P. diazotrophicus*
*BDA59-3* were never detected in the shoots (Figure 3g–o).

### 3.6. The Endophytes Kl. pasteurii BDA134-6 and P. diazotrophicus BDA59-3 Improve N-Fixation in Two Host Rice Species

To evaluate the N-fixing ability of six endophytic strains when they interact with their host plants (*O. glaberrima* RAM133 and *O. sativa* cv. Baldo) an ARA test was carried out on two-week-old plants inoculated with the different strains. Results reported in Figure 4 show that the inoculation of the two *Oryza* species with the strains *Kl. pasteurii* BDA134-6 and *P. diazotrophicus* BDA59-3 led to the highest nitrogenase activity. The lowest nitrogenase activity was observed when *Enterobacter* sp. BDAM41-2 and *Ko. oryzendophytica* BDA137-1 were used for the inoculation of *O. glaberrima* and *O. sativa* plants, respectively.

### 3.7. N2 Gas Incorporation

In addition to the acetylene reduction assay, which is an indirect method to verify nitrogen fixation of endophytes in plants, we have also designed a ^15^N dinitrogen incorporation experiment to follow ^15^N2 gas into rice plants through the endophytic strains *Ko. pseudosacchari* BDA62-3, *Kl. pasteurii* BDA134-6 and *P. diazotrophicus BDA59-3*. When two-week-old plants were analyzed a significant (as a value greater than the declared measurement uncertainty) ^15^N incorporation from ^15^N gas was detected for rice plants inoculated with *P. diazotrophicus BDA59-3* (δ^15^N = 8.9 ± 0.2) as compared to uninoculated ones (δ^15^N = 6.9 ± 0.2). For plants inoculated with *Ko. pseudosacchari* BDA62-3 (δ^15^N = 7.1 ± 0.2) and *Kl. pasteurii* BDA134-6 (δ^15^N = 7.2 ± 0.6) the ^15^N incorporation was not statistically significant. This result indicated that the strain *P. diazotrophicus* BDA59-3 transferred the fixed nitrogen to *O. sativa* plants.

### 3.8. Kl. pasteurii BDA134-6 and P. diazotrophicus BDA59-3 Increase Carbon (C), Nitrogen (N) and Chlorophyll (Chl) Content of Inoculated O. sativa Plants

To analyse the effect of the inoculation with the strains *E. sacchari* BDA62-3, *Kl. pasteurii* BDA134-6 and *P. diazotrophicus* BDA59-3 on the growth of *O. sativa* L. cv. Baldo plants the total content of carbon (C), nitrogen (N) and chlorophyll (Chl) were measured under hydroponic conditions at 14 and 40 DAI. Uninoculated plants were used as reference (Figure 5). A significant increase in all three parameters was measured for plants inoculated with the strain *P. diazotrophicus.* BDA59-3 at both 14 and 40 DAI. When ‘Baldo’ plants were inoculated with *Kl. pasteurii* BDA134-6, the C and N content increased only at 40 DAI, and the measured levels were lower than the ones estimated for plants inoculated with *P. diazotrophicus* BDA59-3. Plants inoculated with the strain *E. sacchari* BDA62-3 did not show significant changes in any of the analysed parameters both at 14 and 40 DAI.

### 3.9. Inoculation of Rice Plants with Kl. pasteurii BDA134-6 Allows Maintenance of the Nitrogenase Function and Increases the Activity of Antioxidative Enzymes under Salinity Stress

‘Baldo’ rice was single-inoculated with the endophytes *Kl. pasteurii* BDA134-6 (good IAA-producer and N-fixer) and *P. diazotrophicus* BDA59-3 (weak IAA-producer and N-fixer) and subjected to salinity stress at 21 DAI. The effect of NaCl treatment on N-fixation and stress response was then evaluated in inoculated plants by using the unstressed plants as reference. After salinity stress, plants inoculated with *Kl. pasteurii* BDA134-6 showed only a slight repression of nitrogenase activity (Figure 6). Instead, for *P. diazotrophicus* BDA59-3-inoculated plants, a significant decrease of nitrogenase activity was estimated (Figure 6). Consistently, the activity of all antioxidative enzymes was significantly enhanced in the tissues of plants inoculated with *Kl. pasteurii* BDA134-6 (Figure 7). For *P. diazotrophicus* BDA59-3-inoculated plants, a positive effect was observed only for the activity of APX (Figure 7).

## 4. Discussion

Because of their domestication, the human interventions in agriculture and the change in environmental conditions plants might have lost their ability to symbiotically associate with endophytic microbes supporting their response against biotic and abiotic stresses [51,52].

In our study, we tested the ability of the stable crop Asiatic rice *O. sativa* L. to exploit the PGP and stress alleviation properties of *O. glaberrima* bacterial endophytes. For this aim, *O. sativa* cv. ‘Baldo’ rice, extensively cultivated in Northern Italy (with a range of temperatures 0–16 °C in winter and 5–29 °C in summer), was inoculated with bacterial endophytes isolated from African rice *O. glaberrima* plants cultivated in the Niger River valley (with a range of temperatures 18–27 °C in winter and 30–40 °C in summer). This region is characterized by degraded soil quality and severe environmental stresses, such as heat and extreme variation in the water regime.

*O. sativa* and *O. glaberrima* are both domesticated rice strains, and *O. glaberrima* has far lower productivity compared to *O. sativa.* However, *O. glaberrima* is cultivated with poor or no agronomic inputs, it has acquired and developed tolerance and resistance against the majority of the biotic and abiotic stresses affecting rice cultivation in Sub-Saharan Africa, and is probably more prone to associate with endophytes promoting plant growth.

The most common endophytic orders detected in this work were the Enterobacterales and Pseudomonadales followed by Xanthomonadales and Bacillales. These results are consistent with other studies concerning the isolation of endophytic bacterial communities from host plants by culture-based methods [53,54] and with the bacterial diversity pattern previously reported for rice microbiome in other studies [55,56]. 

It is well known that most of the endophytic bacteria possess features that are directly or indirectly involved in promoting plant growth [57]. For instance, the production of indole-3-acetic acid (IAA) is an important trait through which endophytic bacteria communicate with the host plants enhancing their growth [58]. IAA is the main auxin acting as phytohormone in many plant developmental processes [59]. Auxin homeostasis has a major impact on root architecture, thereby improving nutrient acquisition [60]. A high percentage of endophytic bacteria isolated in this study produced IAA. This result is in agreement with those reported in the literature, reinforcing the concept that the production of IAA is very common among bacteria with endophytic behavior [61,62]. 

Another important feature of PGPB is the ability to fix nitrogen. Endophytic bacteria can supply fixed atmospheric nitrogen to their host plant by expressing the nitrogenase, a highly conserved enzymatic complex. These bacteria are commonly known as endophytic diazotrophs. Although, endophytic diazotrophs are less efficient in nitrogen-fixation ability than the legume symbiont rhizobia, some N-fixing strains have been identified [58]. In this study, six nitrogen-fixing strains having the *nifH* gene were identified: *Ko. oryzendophytica* (BDA137-1), *E. sacchari* (BDA86-11), *Ko. pseudosacchari* (BDA62-3), *Kl. pasteurii* (BDA134-6), *Enterobacter* sp. (BDAM41-2) and *P. diazotrophicus* (BDA59-3). The presence of the full gene set for nitrogen fixation was confirmed by genome sequencing of three of them (*Ko. pseudosacchari* BDA62-3, *Kl. pasteurii* BDA134-6, and *P. diazotrophicus* BDA59-3). The diazotrophic nature of these endophytes was confirmed by the ARA test performed on free-living strains. The strain *Kl. pasteurii* BDA134-6 had the highest nitrogenase activity. As the nitrogenase enzyme complex in free-living diazotrophs is highly sensitive to molecular oxygen, we speculate that in *Kl. pasteurii* BDA134-6, as well as in the other five isolated diazotrophs, the activity of largely uncoupled respiratory electron transport systems may have protected the nitrogenase enzyme from oxygen damage, as already reported in other studies [63]. Further studies will provide additional data to test such hypothesis.

In the present work, the marker gene *acdS*, which codes for the 1-aminocyclopropane-1-carboxylate (ACC) deaminase, was also identified in six strains belonging to the families of the Ralstoniaceae, Microbacteriaceae, Enterobacteriaceae and Xanthomonadaceae. The presence of *acdS* positive strains in the endosphere of *O. glaberrima* is consistent with its tolerance and resistance against biotic and abiotic stresses. Indeed, lowering ethylene levels by 1-aminocyclopropane-1-carboxylate (ACC) deaminase is considered one of the major mechanisms employed by PGPB to promote plant growth under stress conditions [10]. When the nitrogen-fixing abilities of the six N-fixing strains were evaluated in the host plants *O. glaberrima* RAM133 and *O. sativa* cv. Baldo, the highest nitrogenase activity, was recorded for plants inoculated with the strains *Kl. pasteurii* BDA134-6 and *P. diazotrophicus* BDA59-3, for both rice species. It was also found that these strains were able to colonize *O. sativa* cv. Baldo plants quite efficiently, although the CFU number measured was 10 times lower than the one registered for *O. glaberrima* plants inoculated with the same strains. The higher number of CFU measured for *O. glaberrima* inoculated plants might partially explain the higher ARA efficiency measured for these plants as compared to *O. sativa* inoculated with the same strain. 

The different colonization efficiency observed for the two genetically distant rice species (*O. glaberrima* and *O. sativa*) used in this study is not trivial considering that there are contrasting reports about the host specificity of endophytes. Some researchers have reported that endophytes are only able to promote growth of plants closely related to their natural hosts. On the other hand, there are studies concerning the ability of endophytes to promote growth of diverse host plants [58]. Host facthat, as already observed for legume plants tors such as genotype and host phylogenetic relationships play a role in shaping endophytic communities of rice cultivars. Furthermore, it is increasingly evident that environmental conditions such as soil nutrition, moisture, and temperature have a direct influence on endophere communities [64]. *O. glaberrima* and *O. sativa* have adapted to growth in very different climatic conditions: *O. glaberrima* is adapted to harsh African climate and presents resistance or tolerance to viruses, nematode, bacteria, drought, iron toxicity and high salinity [65,66], while *O. sativa* cv. Baldo plants are grown through a submersion technique in regions of Italy characterized by temperate climate with more than 30 °C of excursion between winter and summer. When the nitrogen-fixing strains *Ko. pseudosacchari* BDA62-3 (weak IAA-producer and N-fixer) and *Kl. pasteurii* BDA134-6 (good IAA-producer and N-fixer) and *P. diazotrophicus* BDA59-3 (weak IAA-producer and N-fixer) were used to inoculate *O. sativa* cv. Baldo we verified that none of them was able to significantly colonize the shoot tissues. By contrast, when the roots were analysed, their colonization efficiency was in the following order: *Kl. pasteurii* BDA134-6 > *Ko. pseudosacchari* BDA62-3 > *P. diazotrophicus* BDA59-3.

Fluorescence microscopy visualization confirmed the ability of *Ko. pseudosacchari* BDA62-3, *Kl. pasteurii* BDA134-6, and *P. diazotrophicus* BDA59-3 to colonize the rice root endosphere. For the strain *Kl. pasteurii* BDA134-6, the internal colonization of root hairs could be linked to the presence of a complete set for the T6SS in the genome of this strain, as the genomic sequence suggested. Indeed, it has been shown that a T6SS mutant of *Kosakonia* sp. KO348 displayed a significant decrease in rice rhizoplane and root endosphere colonization, thus suggesting a role in the host-bacteria colonization or interaction [67]. When the content of C, N and chlorophyll was measured for ‘Baldo’ plants inoculated with the three selected strains, an increase in all three parameters was observed for *P. diazotrophicus* BDA59-3 both at 14 and 40 DAI. These results, together with the data obtained for the nitrogenase activity and the ^15^N incorporation, clearly demonstrated that the nitrogen fixed by the strain *P. diazotrophicus* BDA59-3 was directly transferred to *O. sativa* plants from their early stages of growth. 

On the other hand, for plants inoculated with *Kl. pasteurii* BDA134-6, the level of N and C content increased only at 40 DAI. We hypothesize that for the strain *Kl. pasteurii* BDA134-6 the transfer of fixed N to the host plant occurs in a later phase of plant growth as compared to *P. diazotrophicus* BDA59-3. 

Kinetic studies are required to fully evaluate the accumulation of fixed N in plant tissues over time. The highest nitrogenase activity observed for plants inoculated with *Kl. pasteurii* BDA134-6, which is at the same time a N-fixer and IAA-producer, was consistent with previously published data concerning the positive effect of IAA on nitrogen fixation of rhizobacteria interacting with their host plants [35,38]. Furthermore, the levels of C, N, and Chl measured for plants inoculated with the strain *Ko. pseudosacchari* BDA62-3 were consistent with the low nitrogenase activity and the negligible ^15^N incorporation measured for this strain. We thus hypothesize that, as already observed for legume plants [68], the nitrogen fixation by endophytes within rice plant tissues is tightly linked to carbon metabolism. 

Finally, considering that rice is one of the most salt-sensitive cereals and that the anti-oxidative enzymes scavenge ROS produced as metabolic by-products during osmotic stress and convert them into non-reactive form, the activity of the enzymes APX, CAT, POX and SOD was measured for ‘Baldo’ plants inoculated with *Kl. pasteurii* BDA134-6 and *P. diazotrophicus* BDA59-3 under salt stress conditions. 

Our study shows that the strain *Kl. pasteurii* BDA134-6 performed significantly better in the enhancement of all anti-oxidative enzymes activities compared to strain *P. diazotrophicus* BDA59-3. Consistent with these results, a mild loss of nitrogenase activity was observed for *Kl. pasteurii* BDA134-6-inoculated plants, while a stronger reduction was registered for *P. diazotrophicus* BDA59-3-inoculated ones. We speculate that the positive results observed for the strain *Kl. pasteurii* BDA134-6 could be linked to its ability to produce higher IAA levels. Indeed, the increase in the level of this endogenous molecule could lead, as previously observed for rhizobia strains [39], to alterations in the levels of other important hormones, such as ethylene, which promotes fruit ripening and leaf senescence under abiotic stress. Finally, even if *P. diazotrophicus* BDA59-3 and *K. pasteurii* BDA134-6 belong to species for which pathogenic strains are known [69] and potential toxin genes were found in their genomes, the positive effects observed for plants inoculated with *Kl. pasteurii* BDA134-6 reinforce the suggestion that the bacterial nanomachine T6SS provides fitness and colonization advantage *in planta* and that its role is not restricted to virulence [70].

## 5. Conclusions

Our study demonstrated that the culturable endophytes isolated from the African rice *O. glaberrima*, adapted to the harsh environments of West Africa (Mali), were able to colonize quite efficiently the domesticated Asian rice *O. sativa* cv. Baldo, selected to growth in the typical cold and wet environmental conditions of Northern Italy. Our results showed that the inoculation of *O. sativa* cv. Baldo plants with the nitrogen-fixing endophyte *P. diazotrophicus* BDA59-3 positively affected its N-fixation and the content of N, C and chlorophyll. These results support the hypothesis that the efficient endophytic colonizers with good ability to fix N stimulate the C and N metabolism in the host plant in a coordinated manner [71].

Our results also revealed that, under salinity stress, ‘Baldo’ plants inoculated with the strain *Kl. pasteurii* BDA134-6 showed higher activity of enzymes involved in the stress response. We speculate that this effect could be directly or indirectly related to the production of IAA and to the N-fixing activity observed for *Kl. pasteurii* BDA134-6. Our findings suggest that the recovery of microbial biodiversity from compatible plants grown with minimal human intervention and without the addition of nitrogen fertilizers and its introduction into commercial crops, perhaps as seed treatments, could be an effective strategy to reduce reliance on chemical fertilizers and to enhance the plant’s stress response. 

For the success of these approaches, it is important to consider that the new inoculated endophytes must balance with the resident microbiome of the domesticated crops, and this might require a few rounds of open field selection. Furthermore, the evaluation of the effects related to the use of any opportunistic pathogens is another important aspect that should be considered before proceeding with open field experiments.

## Figures and Tables

**Figure 1 microorganisms-09-01714-f001:**
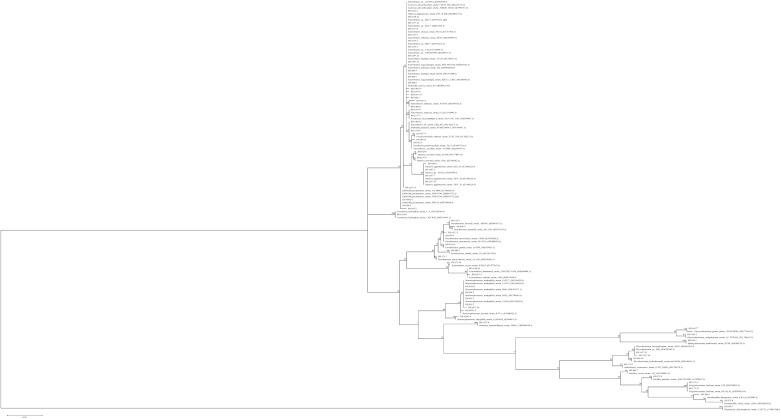
Phylogenetic tree based on the 16S rRNA gene sequences corresponding to endophytic strains inhabiting rice tissues. The sequences for closely related strains were recovered from GenBank and included in the tree. The accession numbers of these strains are shown in parenthesis.

**Figure 2 microorganisms-09-01714-f002:**
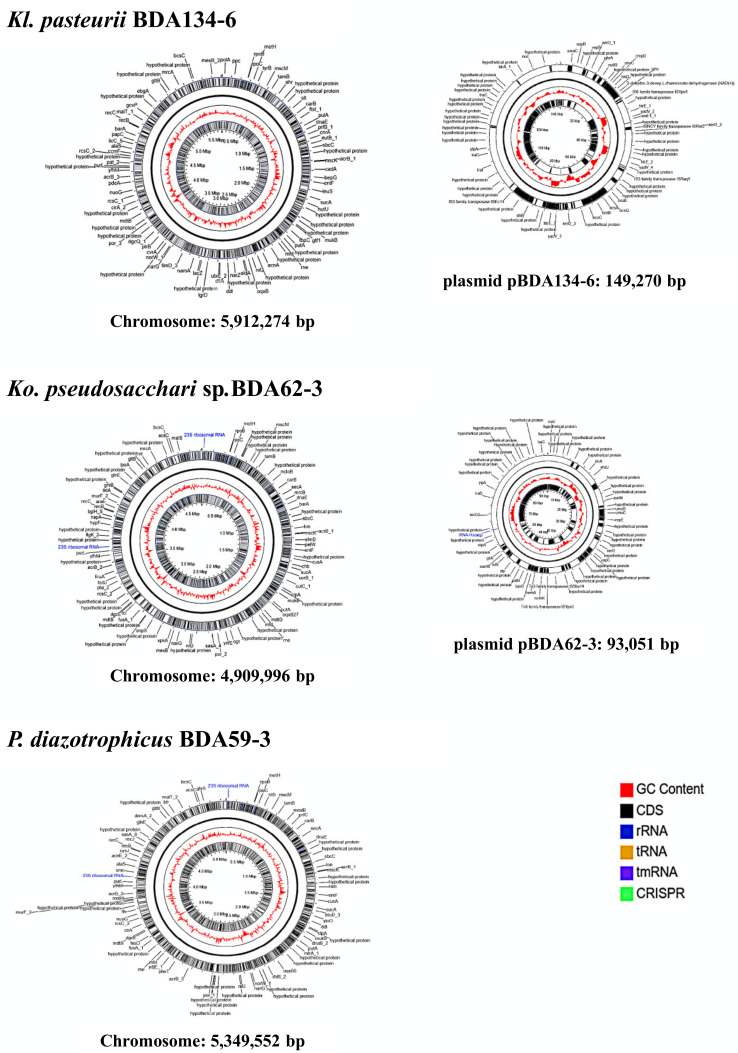
Circular maps of the genomes of *Kl. pasteurii* BDA134-6, *Ko pseudosacchari* BDA62-3 and *P. diazotrophicus* BDA59-3 showing the inner GC content and annotation. Maps were obtained from the CGView Server (http://cgview.ca/, https://doi.org/10.1093/bib/bbx081, accessed on 1 April 2021).

**Figure 3 microorganisms-09-01714-f003:**
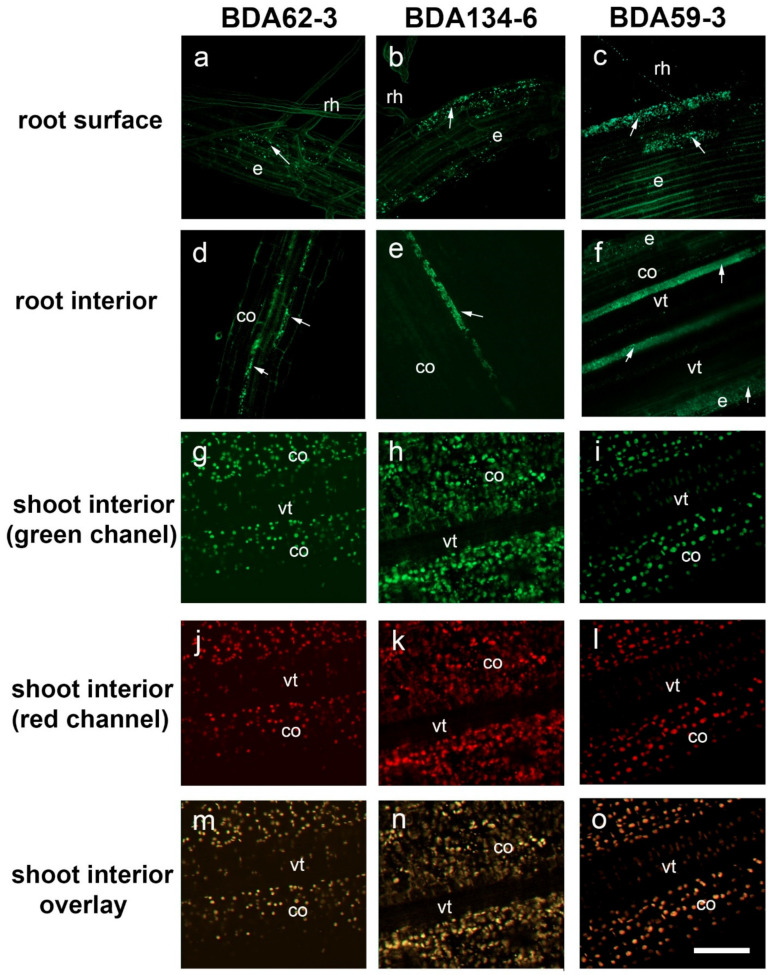
Visualization of the tagged strains *Ko. pseudosacchari* BDA62-3, *Kl. pasteurii* BDA134-6 and *P. diazotrophicus* BDA59-3 in *O. sativa* root and shoot tissues using confocal laser scanning microscopy (CLSM). Confocal analysis was performed on whole representative roots (**a**–**f**) and shoots (**g**–**o**) to show surface and inner colonization of the tagged bacterial strains. Images are projections of 10 optical sections for all panels except for panels (**g**–**o**) where projections were made of 20 optical sections. The focal step size was 4 µm. (**a**–**c**) Detection of BDA62-3, BDA134-6 and BDA59-3 bacteria on the root surface and internal colonization of epidermal cells, including the modified epidermal cells (root hair cells). (**d**–**f**) The internal colonization of the intercellular spaces in the cortex and vascular tissues of the root differentiation zone. The interior of the vascular tissue was never observed invaded by any of the different bacterial strains. (**g**–**o**) Images of the shoots’ interior showing the cortex and the vascular tissues. (**g**–**i**) Bacteria were never detected in the shoots but chloroplasts, which are highly autofluorescent, were also detected using both the green (**g**–**o**) and the red channel (**j**–**l**). (**m**–**o**) Overlay of green and red panels of the rice shoots for the three bacteria strains showing overlay of the fluorescent signals for chloroplasts. No bacteria were visualized in green in any case. Bacteria colonization is arrowed. co, cortical cells; e, epidermal cells; rh, root hairs; vt, vascular tissue. Scale bar represents 50 µm.

**Figure 4 microorganisms-09-01714-f004:**
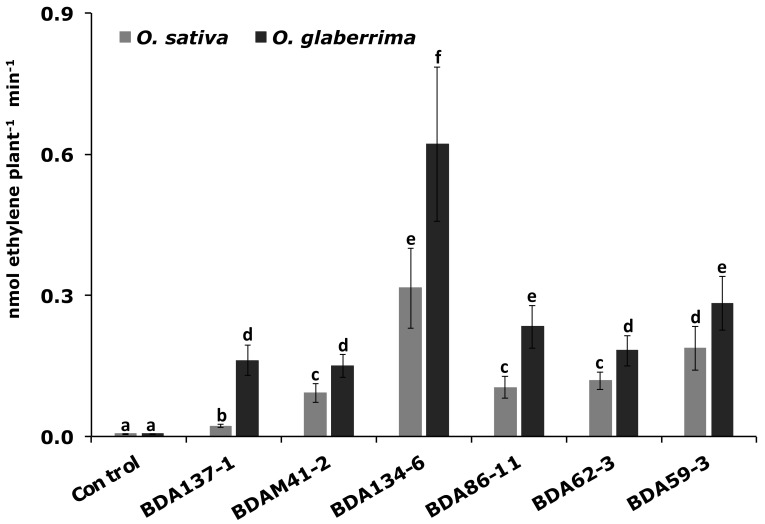
Nitrogenase activity in *O. glaberrima* RAM133 and *O. sativa* L. cv. Baldo plants inoculated with N-fixing endophytes. Two-week-old rice plants of primary (*O. glaberrima*) and secondary (*Oryza sativa*) inoculated host were carefully removed from the pots and used for the ARA test. Data are the means ± SD of at least six independent biological replicates each carried out at different times. Means with different letters are significantly different at 5% level of confidence (Tukey’s post hoc test).

**Figure 5 microorganisms-09-01714-f005:**
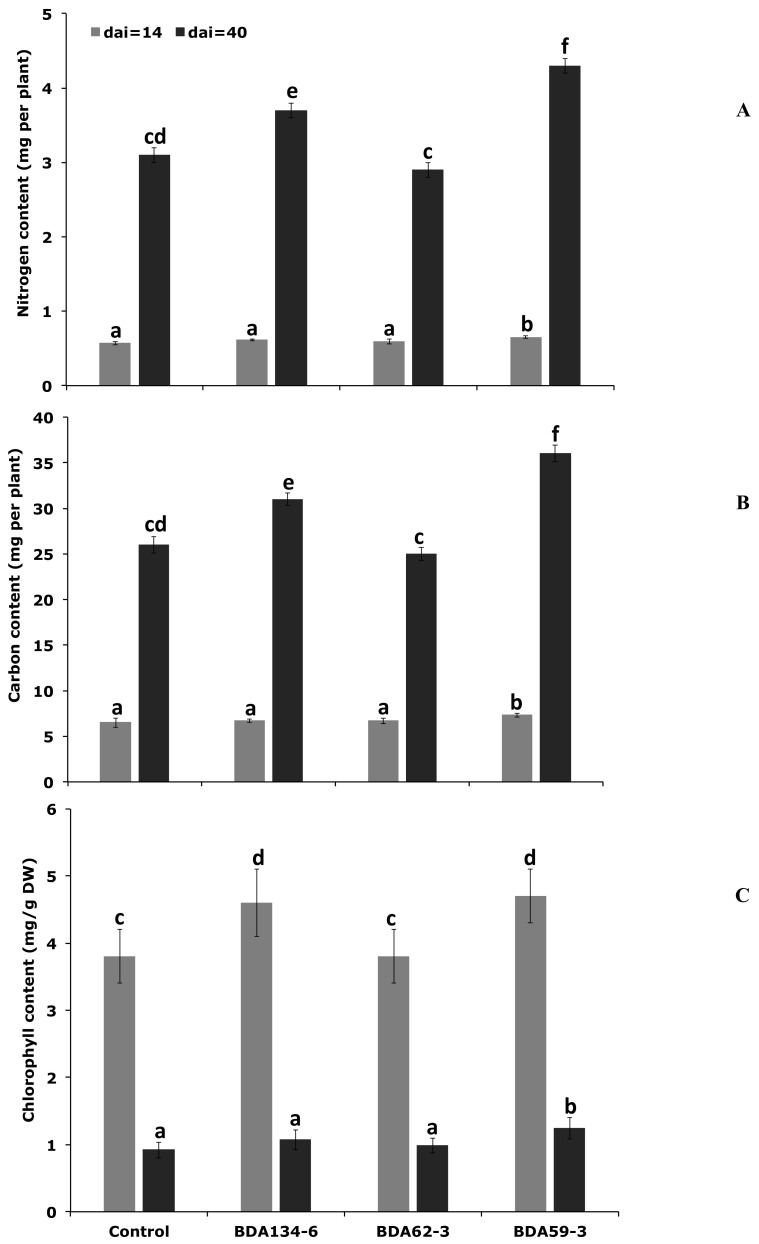
Effect of inoculation with *Ko. pseudosacchari* BDA62-3, *Kl. pasteurii* BDA134-6 and *P. diazotrophicus* BDA59-3 on physiological parameters of *O. sativa* L. cv. Baldo plants. *O. sativa* plants were inoculated with the individual strains and grown in hydroponic conditions as described in Material and Methods section. At 14 and 40 DAI, plants were removed from the growth unit and the leaves detached and used for the analyses of N (**A**), C (**B**), and chlorophyll (**C**) content. Data are the means ± SD of at least five independent biological replicates each carried out at different times. Means with different letters are significantly different at 5% level of confidence (Tukey’s post hoc test).

**Figure 6 microorganisms-09-01714-f006:**
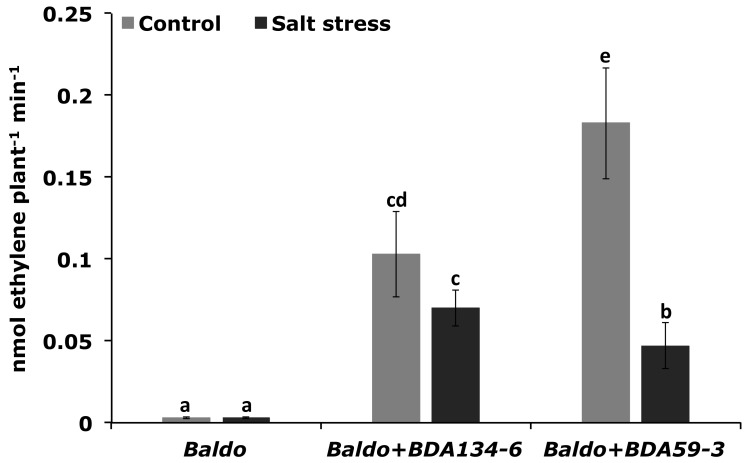
Effect of salinity stress on nitrogenase activity of *O. sativa* L. cv. Baldo plants inoculated with the endophytes *Kl. pasteurii* BDA134-6 and *P. diazotrophicus* BDA59-3. Three-week-old rice plants inoculated with the selected strains were carefully removed from the pots, treated with 0.3 M NaCl for 24 h and used for the ARA test. Data are the means ±SD of at least six independent biological replicates, each carried out at different times. Means with different letters are significantly different at 5% level of confidence (Tukey’s post-hoc test).

**Figure 7 microorganisms-09-01714-f007:**
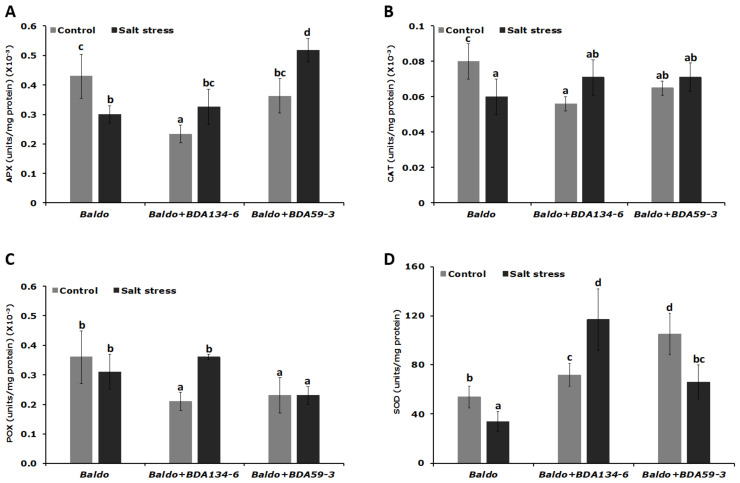
Impact of salinity stress on the activity of the anti-oxidative enzyme in *Kl. pasteurii* BDA134-6- and *P. diazotrophicus* BDA59-3-inoculated plants. Three-week-old rice plants inoculated with the strains *Kl. pasteurii* BDA134-6 and *P. diazotrophicus* BDA59-3 were treated with 0.3 M NaCl for 24 h. After NaCl treatment, the soluble proteins were extracted from leaves and the activity of the enzymes ascorbate peroxidase (APX) (**A**), catalase (CAT) (**B**), peroxidase (POX) (**C**) and superoxide dismutase (SOD) (**D**) measured. Data are the means ± SD of at least five independent biological replicates each carried out at different times. Means with different letters are significantly different at 5% level of confidence (Tukey’s post-hoc test).

**Table 1 microorganisms-09-01714-t001:** Nitrogenase activity and IAA production in bacterial cultures of N-fixing endophytes isolated from *O. glaberrima.*

Strain	Nitrogenase Activity ^§^ (nmol Ethylene/mg Cells/min)	IAA (nmol/mg Cells) ^ξ^
*Ko. oryzendophytica BDA137-1*	0.032 ± 0.004 ^c^	8.7 ± 1.4 ^c^
*Enterobacter* sp. *BDAM41-2*	0.025 ± 0.005 ^c^	6.1 ± 0.3 ^c^
*Kl. pasteurii BDA134-6*	0.160 ± 0.024 ^a^	31 ± 2 ^b^
*E. sacchari BDA86-11*	0.064 ± 0.008 ^b^	7.4 ± 0.3 ^c^
*Ko pseudosacchari BDA62-3*	0.049 ± 0.008 ^b^	7.7 ± 0.5 ^c^
*P. diazotrophicus BDA59-3*	0.044 ± 0.003 ^b^	61 ± 10 ^a^

**^§^** Bacterial cells grown in M9 minimal ammonium chloride medium were suspended in the same medium without nitrogen source and incubated under a hypoxic atmosphere (2% O2) and 10% acetylene. After 18 h of incubation at 30 °C the amount of the ethylene produced was measured by using a gas chromatograph. **^ξ^** Bacterial cells were grown overnight in LB medium containing 100 μM L-tryptophan at 30 °C. The culture supernatant was then mixed with the Salkowski reagent and incubated for 30 min in the dark at room temperature. The absorbance of the mixtures was then estimated at 530 nm. The bacterial cells were dried and weighed for data normalization. The values reported in the table are the mean ± SD of at least five independent bacterial cultures each grown at different times and from different bacterial colonies. Means with different letters are significantly different at 5% level of confidence.

**Table 2 microorganisms-09-01714-t002:** Identification of *nif* gene cluster in the genome of *Kl. pasteurii* BDA134-6, *Ko. pseudosacchari* BDA62-3 and *P. diazotrophicus* BDA59-3.

Gene Code	Gene Name
*Kl. pasteurii BDA134-6*	
BDA134-6_02421	*nifJ*
BDA134-6_03551	*nifH*
BDA134-6_03552	*nifD*
BDA134-6_03553	*nifK_1*
BDA134-6_03556	*nifD_2*
BDA134-6_03557	*nifK_2*
BDA134-6_03560	*nifS*
BDA134-6_03562	*nifW*
BDA134-6_03565	*nifF*
BDA134-6_03566	*nifL*
BDA134-6_03567	*nifA*
BDA134-6_03568	*nifB*
*Ko. pseudosacchari* BDA62-3	
Entero_BDA62.3_02197	*nifQ*
Entero_BDA62.3_02198	*nifB*
Entero_BDA62.3_02199	*nifA*
Entero_BDA62.3_02200	*nifL*
Entero_BDA62.3_02201	*nifF*
Entero_BDA62.3_02203	*nifZ*
Entero_BDA62.3_02204	*nifW*
Entero_BDA62.3_02206	*nifS*
Entero_BDA62.3_02207	*nifU*
Entero_BDA62.3_02208	Dinitrogenase iron-molybdenum cofactor
Entero_BDA62.3_02209	*nifK_1*
Entero_BDA62.3_02210	*nifD_1*
Entero_BDA62.3_02211	Dinitrogenase iron-molybdenum cofactor
Entero_BDA62.3_02212	*nifT/fixU*
Entero_BDA62.3_02213	*niK_2*
Entero_BDA62.3_02214	*nifD_2*
Entero_BDA62.3_02215	*nifH*
*P. diazotrophicus* BDA59-3	
Citro_BDA59-3_02396	*nifJ_1*
Citro_BDA59-3_02623	*nifJ_2*
Citro_BDA59-3_02624	*nifH*
Citro_BDA59-3_02625	*nifD_1*
Citro_BDA59-3_02626	*nifK_1*
Citro_BDA59-3_02627	*nifT/fixU*
Citro_BDA59-3_02628	*Dinitrogenase iron-molybdenum cofactor*
Citro_BDA59-3_02629	*nifD_2*
Citro_BDA59-3_02630	*nifK_2*
Citro_BDA59-3_02631	*Dinitrogenase iron-molybdenum cofactor*
Citro_BDA59-3_02632	*nifU*
Citro_BDA59-3_02633	*nifS*
Citro_BDA59-3_02635	*nifW*
Citro_BDA59-3_02636	*nifZ*
Citro_BDA59-3_02638	*nifF*
Citro_BDA59-3_02639	*nifL*
Citro_BDA59-3_02640	*nifA*
Citro_BDA59-3_02641	*nifB*

**Table 3 microorganisms-09-01714-t003:** Endophytic colonization of *O. glaberrima* and *O. sativa* plants by six N-fixing endophytes.

Strain	*O. glaberrima* (CFU per Plant) ^§^ (×10 ^5^)	*O. sativa* (CFU per Plant) ^§^ (×10 ^5^)
*Ko. oryzendophytica BDA137-1*	151 ± 24 ^e^	10 ± 2 ^d^
*Enterobacter* sp. *BDAM41-2*	2264 ± 410 ^a^	10 ± 2 ^d^
*Kl. pasteuri BDA134-6*	1064 ± 93 ^b^	176 ± 39 ^a^
*E. sacchari BDA86-11*	613 ± 76 ^c^	180 ± 38 ^a^
*Ko. pseudosacchari BDA62-3*	312 ± 92 ^d^	154 ± 24 ^b^
*P. diazotrophicus BDA59-3*	384 ± 57 ^d^	49 ± 8 ^c^

**^§^** The root of germinated seeds of *O. glaberrima* and *O. sativa* were cut (0.5 cm from the bottom) with a sterile bistoury and incubated in Petri dishes with 50 mL of 1× PBS solution containing each strain to a final concentration of 10 ^6^ cells mL^−1^ for 4 h at room temperature. Inoculated seeds were grown in sand perlite soil. At 14 days after the inoculation, the endophytes were re-isolated from the root tissues to evaluate the number of CFU per plant. The values reported in the table are the mean ± SD of at least five independent replicates each carried out at different times. Means with different letters are significantly different at 5% level of confidence.

## Data Availability

The 16S rRNA gene partial sequences were deposited at NCBI GenBank with the following accession numbers (see also Appendix A): MT241164 through to MT241232. Genome sequencing is deposited under Bioproject (PRJNA670042).

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
