# Peer review of "Endophytes from African Rice (Oryza glaberrima L.) Efficiently Colonize Asian Rice (Oryza sativa L.) Stimulating the Activity of Its Antioxidant Enzymes and Increasing the Content of Nitrogen, Carbon, and Chlorophyll"

_microorganisms, 2021, doi:10.3390/microorganisms9081714_

Round 1

Reviewer 1 Report

This article aims to isolate rice endophytic bacteria from tissue of African rice, then characterize their PGP traits, analyze the ability of Asian rice colonization including their effect of rice growth promotion and evaluate the response of inoculated rice against salinity condition.

Generally, the techniques used in this study are very nice. However some results can be further evaluated in term of function of endophytic bacteria and plant response in future. There are some few concerns as followed,

  1. Along with the objective in No.4, authors want to evaluate the response of inoculated plants to abiotic stress conditions. But this study only conducted in salinity. I would recommend to modify this objective.
  2. There are too many references in this MS. Author should reduce and select only very critical and/or more up to date.
  3. In Fig 5, 6 and 7, the letters showing the statistically difference on the top of each bar are strange. Please recheck whether are correct. If they are correct the description how to calculate must be amended in the figure legends.
  4. In discussion, author showed the different host preference of isolated endophytes. There are some reports discuss that T3SS has involved in this situation. Please reconsider and add into this part. There are also some reports demonstrated the different host preference of endophyte bacteria among O. sativa/ O. indica. Please also reconsider this point for discussion.
  5. The relationships of IAA production, ACC-deaminase production and N2-fixation are not always directly provided from endophytic bacteria to plant growth. I would recommend to read the article written by T. Greetatorn et al., 2019 in Letter in Applied Microbiology. Perhaps author can make discussion part more solid.

There are also some minor concerns:

  1. After using full name of scientific name at first, next time should be abbreviated for example Oryza glaberrium, next should be O. glaberrium. This is also the same as Kasakonia and Klebsiella should be Ka. and Kl. . Please check through at this MS.
  2. In keywords: “physiological parameters” should change to be “rice endophytic bacteria” instead.
  3. In introduction the first paragraph “It is widely…” till line No. 13….[7-9]” can be deleted.
  4. In 2.10.1 after (ARA) “which is…..N2 gas” can be deleted.
  5. In 3.8 “didn’t” should be “did not”
  6. In discussion, the first paragraph can be deleted.

Author Response

We really appreciate the Reviewer’ contribution to improve the whole manuscript. In the revised version of the manuscript we addressed all the criticisms, concerns and recommendations raised by the Reviewer. 

This article aims to isolate rice endophytic bacteria from tissue of African rice, then characterize their PGP traits, analyze the ability of Asian rice colonization including their effect of rice growth promotion and evaluate the response of inoculated rice against salinity condition.

Generally, the techniques used in this study are very nice. However some results can be further evaluated in term of function of endophytic bacteria and plant response in future.

There are some few concerns as followed,

  1. Along with the objective in No.4, authors want to evaluate the response of inoculated plants to abiotic stress conditions. But this study only conducted in salinity. I would recommend tomodify this objective.

Authors’ reply - We have considered your suggestion and modified the text accordingly.

  1. There are too many references in this MS. Author should reduce and select only very critical and/or more up to date.

Authors’ reply - The revised manuscript contains 30% fewer References. We have removed the older and the less essential ones.

  1. In Fig 5, 6 and 7, the letters showing the statistically difference on the top of each bar are strange. Please recheck whether are correct. If they are correct the description how to calculate must be amended in the figure legends.

Authors’ reply - We have verified that in the elaboration of the Figures we had made some mistakes. The histogram bars in the revised Figures have been labeled differently.

  1. In discussion, author showed the different host preference of isolated endophytes. There are some reports discuss that T3SS has involved in this situation. Please reconsider and add into this part. There are also some reports demonstrated the different host preference of endophyte bacteria among
  2. sativa/O. indica. Please also reconsider this point for discussion.

Authors’ reply – When the genome sequences of BDA134-6, BDA62-3 and bBDA59-3 were analysed the type III secretion system was retrieved only for K. pseudosacchari BDA62-3. However, no effector proteins were identified. The importance of the hosts in determining the microbial communities of rice cultivars was highlighted in the Discussion of the revised paper.

  1. The relationships of IAA production, ACC-­deaminase production and N -­fixation are not always directly provided from endophytic bacteria to plant growth. I would recommend to read the article written by T. Greetatorn et al., 2019 in Letter in Applied Microbiology. Perhaps author can make discussion part more solid.

Authors’ reply – This issue was addressed in the Discussion of the revised paper. We have also cited the suggested Reference.

There are also some minor concerns:

  1. After using full name of scientific name at first, next time should be abbreviated for example , Oryza glaberrima, next should be O. glaberrima. This is also the same as Kosakonia and Klebsiella should be Ko. and Kl. . Please check through at this MS.

Authors’ reply – We checked the entire manuscript and revised the nomenclature of all mentioned strains.

  1. In keywords: “physiological parameters” should change to be “rice endophytic bacteria” instead.

Authors’ reply – We followed your suggestion and modified the Keywords accordingly.

  1. In introduction the first paragraph “It is widely…” till line No.13….[7-­9]” can be deleted.

Authors’ reply – The Introducion of the revised manuscript does not contain this paragraph.

  1. In 2.10.1 after (ARA) “which is…..N gas” can be deleted.

Authors’ reply – We removed this concept fro the ARA assay SubSection.

  1. In 3.8 “didn’t” should be “did not”

Authors’ reply – The text has been changed in the revised manuscript.

  1. In discussion, the first paragraph can be deleted.

Authors’ reply – Further to your suggestion we modified the text in the Discussion section: some paragraphs have been deleted while others, suggested by the reviewer, have been added.

Reviewer 2 Report

See the comments in the attached file

Author Response

We really appreciate the Reviewer’ contribution to improve the whole manuscript. In the revised version of the manuscript we addressed all the criticisms, concerns and recommendations raised by the Reviewers. We introduced new data into Supplementary Materials as Tables.

  1. KY927406.1 is a Pantoea strain not a Curtobacterium (gram positive actinobacteria). https://www.ncbi.nlm.nih.gov/nuccore/KY927406.1/

Please, carefully check if there are other misannotations and highlight the 16S rRNA belonging to type strains

Authors’ reply – We acknowledge the reviewer for her/his kind and precise checking. We did a careful check of all strains reported and found no other misnaming. KY927406 was removed from the dataset since the record KY927406.1 indicates Curtobacterium in the header but Pantoea in the taxonomy, questioning about the reliability of this sequence record.

  1. After a taxonomy check using the Type (Strain) Genome Server (Nat. Commun. 10, 2182 (2019)), BDA59-3 is a Phytobacter diazotrophicus.

I highly recomend a phylogenetic characterization of these strains using a phylogenomic-based approach instead of relaying only on the analysis of the 16S rRNA

Authors’ reply – We agree with the reviewer, and we thank for the suggestion. We have included the results of runs on Type (Strain) Genome Server for the three genomes reported (A Supplemental file with genome matches is now reported) and amended the naming of Citrobacter sp. BDA59-3 to Phytobacter diazotrophicus BDA59-3

  1. One of the genome annotation files is missing in Folder S1

Authors’ reply – all genome annotation files are now present (.gff and .faa for each of the three genomes)

  1. I can't find the Figure S1 in the supplementary material folder

Authors’ reply – Figure S1 is now reported.

  1. I can't find in the Material and Methods section which tool have been used to assign protein coding genes to Clusters of Orthologous Genes (COGs)

In addition, kindly provides as Supplementary Material the chromosomal coordinates of the Type III and Type VI gene clusters identified in each strain.

Authors’ reply – We have specified the tool used to COG assignment (Prokka) and coordinates for T3SS and T4SS are now reported in Table S4.

  1. Just like the nif gene cluster, provide the gene content of each cluster as Supplementary material

Authors’ reply – We are now reporting the information. We have included a novel supplemental information in the form of the dataset of KAAS annotation where KO terms for functional modules are provided for each annotated gene. Table S5 is reporting the presence/absence of each ortholog with respect to KAAS annotation (KO terms) for flagellar biosynthesis and chemotaxis.

  1. The strains characterized in this study are known opportunistic pathogens and also show very high levels of resistance to multiple antibiotics. I think is important to discuss the potential risks of using such strains in the open field

Authors’ reply – We agree with the reviewer. We have now added a paragraph in the conclusion section on the need for careful risk evaluation before inoculant application.

English was checked throughout the manuscript and minor spelling typos were fixed

This manuscript is a resubmission of an earlier submission. The following is a list of the peer review reports and author responses from that submission.

Round 1

Reviewer 1 Report

The manuscript studied culturable, nitrogen-fixing endophytic bacteria isolated from African rice (Oryza glaberrima) which were identified at the genus level based on 16S rRNA sequence analysis and characterized for plant growth promoting (PGP) traits. Sixty-nine isolates were assessed for three common studied PGP traits, such as IAA-production, nitrogen-fixation and presence of the gene acdS.  Six nitrogen-fixing endophytic bacteria were selected and tested for their nitrogen-fixing and colonization ability on the domesticated Asian rice Oryza sativa cv. Baldo. Moreover, the plant growth promoting effects of the selected bacteria were evaluated under normal and salinity stress conditions.

The topic of the manuscript is currently interesting since it deals with the recovery of microbiota from wild plants and the exploitation of selected and characterized PGPB for increasing crop stress resilience to face climate change. The study provides new bacterial strains as candidates to support plant growth. However, in future studies it would be interesting to know whether the studied bacteria do not cause any diseases in other crops in order to be used with safety as field inoculants.

Major comments:

Figure 1: The authors used 16S sequences from strains deposited in the GenBank to construct the 16S phylogeny. However, this approach carries the risk for incorrect phylogenetic affiliation of the studied isolates since several strains in the literature are either misnamed or have uncertain species affiliation. I would suggest rebuilding the 16S phylogeny by including the closest type strains, at least, for the selected six PGPB.

Lines 341-343: the authors checked the presence of the nifH and acdS by performing PCR analysis with specific primers. However, these primers are degenerate and sometimes could produce unspecific DNA products of the same size as the expected ones. In other words, the presence of a DNA band with a size similar to the expected does not prove the presence of the studied gene. Did the authors verify the identity of the PCR products, at least for acdS, by sequence analysis?

Moreover, the authors mention (lines 524-525) that “the strain Stenotrophomonas rhizophila M41-6 was positive for the acdS gene, while other strains within the Stenotrophomonas genus did not”. That amplification might be due to an unspecific PCR product. Did the authors check the identity of that product by sequence analysis?

Table 4: The numerical results should be presented consistently in O. sativa and O. glaberrima. That is, the results for O. sativa should be converted to the 3rd power of 10.

Lines 440-441: Based on the results of Figure 5, the inoculation with BDA59-3 significantly decreased the activity of nitrogenase under salt stress instead of BDA134-6, as authors claimed.

Lines 552-553 : “the colonization efficiency of the strains BDA59-3- and BDA62-3 was very similar, while the strain BDA134-6 was a weaker colonizer”. I think this needs to be rewritten. According to the table 4, the strains BDA134-6 and BDA62-3 could be considered stronger colonizers than BDA59-3 on O. sativa plants. In contrast, the results of table 5 are shown that the strains BDA59-3- and BDA62-3 are stronger colonizers. The authors should discuss this discrepancy of the results presented in tables 4 and 5 and whether the differences could be attributed to different growth conditions, soil-perilte versus hydroponic conditions.

L555-557 : “When the  content of C, N and chlorophyll ………. inoculated with the strain BDA59-3”. According to the results of Figure 4, the content of N and chlorophyll at 13 DAI showed a significant increase in plants inoculated with the strain BDA59-3, while the C content did not.

Minor comments

Line 92: change “expression of the gene acdS” to “presence of the gene acdS”. The authors used genomic DNA to check the presence of the gene and not its expression.

Lines 559-580 : “Our results showed that the nitrogen-fixing endophyte….”. Name the isolate.

Line 580: “...the highest level of IAA positively effects…” change to “…the highest level of IAA positively affects…”

Reviewer 2 Report

Antibiotic resistance was used to quantify endophytic bacterial populations but no information is provided regarding why the specific types and amounts of antibiotics were used. Are these endophytes naturally resistant to these antibiotics or were they genetically modified? This is needs clarification.

The use of Salkowsky's reagent is well know to react with a range of indole compounds so using this reagent to measure IAA specifically is invalid.

The use of the acetylene reduction assay is not reliably quantitative and cannot be used to draw reliable conclusions about the amount of nitrogenase activity in bacterial cultures or plants. It is a quick and easy way to determine if nitrogenase activity is present or absent in a bacterial strain, not how much nitrogenase activity exists. A 15N dilution technique is the proper way to assess N contributions from diazotrophs to plants after inoculation.

Finally, why were rice plants grown for only 2 weeks? This is not long enough to properly evaluate the true effects of bacteria on plant growth and development.

Reviewer 3 Report

Summary of study

The authors aimed at identifying plant endophytes from the African rice species O. glaberrima that reportedly shows higher resistance against salt- and nutrient deficiency stress. Hypothesizing that PGP endophytes play a role in this mechanism, they isolated endophytic strains from rice plants, characterized them regarding their taxonomic identity and nitrogen fixation properties and inoculated the commercially used rice plant O.sativa and O.glaberrima with these strains. Successfully inoculated plants showed lower CFU numbers than the original host, demonstrating the specificity of endophytic relationships. N-fixing activities were also analysed, showing increased rates by some isolates. Furthermore, Oryza sativa L. cv. Baldo was inoculated by two promising isolates and further tested regarding growth, nitrogen fixation rates, nutrient content and salt stress.  The authors report, that inoculated plants showed higher activity of antioxidant enzymes and thus indicating higher resistance to salt stress.

This study emphasizes the importance of isolation and cultivation of isolates and the big potential of cross inoculation of endophytic bacteria as plant growth promoting agents. It further adds valuable information on culturable endophytes from a rice species that is not the commonly used O.sativa. Agricultural applications of microbial agents are likely to be of high importance in the future, making studies like this one an important contribution to this process.

Material and Methods

2.1 Isolation of Endophytic Bacteria from Oryza glaberrima Rice Tissues

Isolation: Why did you cultivate for five days only? Several endophytes or symbionts are very slow growing, leading to cultivation times exceeding a week or two (e.g. root nodule bacteria) – Was there a specific reason for five days?

Line 112-114 why the change of medium?

Line 110: Please clarify: did you pick a specific type of morphology?

2.2.3 Cloning and Sequencing

Why did you subclone the 16S fragment before sequencing? Identification of isolates, also for full length sequencing does not require previous subcloning as in former times, please specify why you chose the more complicated way of previous subcloning prior to 16S sequencing. Please explain further for a better understanding of your strategy.

2.2.4. Phylogenetic characterization

Why MEGA6? I would suggest using a newer version like MEGAX.

2.11. Statistical Analysis

Please specify for what analyses that procedure was applied. Did you check for normal distributions? I would appreciate more explanations for the choice of methods here.

Results

3.1. Identification of Endophytic Bacteria

The accession numbers are not available yet. Please provide a reviewer link for the sequences or release the sequences.

Table 1: I suggest adding more taxonomic information to the table, for better connectivity with the text and figure 2

3.2. PGP-traits of bacterial Endophytes

Table 3: Adding the taxonomic identity would increase readability.

3.7. Inoculation of Rice Plants with Ralstonia picketii BDA134-6 Allows Maintenance of the Nitrogenase 434 Function and Increases the Activity of Antioxidative Enzymes under Salinity Stress

Figure 6: How do you explain the drastically higher peroxidase activity in the not inoculated control?

Discussion

I would suggest splitting and restructuring the discussion into single sections, similar to the results, otherwise the discussion seems to cover all necessary sections.

Specific comments

Line 14: environments

Line 14: Rephrase the sentence, since plants and eenvironmental samples are already used for that purpose.

Line 18: rephrase: Rather “with” than on?

Lines 20-23: in both strains at the same rate? If not, please clarify to avoid misunderstandings

Line 40: harsh

Line 50: please add reference

Lines 61-62: please add references

Line 66: I assume you mean α-Ketobutyrate, please use the alpha symbol

Lines 70-71: add reference

Lines 72-73: rephrase to improve grammar structure and add reference

Lines 104-106: Please rephrase. Avoid “using” for the sterilization process.

Line 253: swap “produced” and “ethylene”

Line 278: the linestart is tabbed out

Line 465: I suggest “biodiversity of rhizosphere microbiomes”

Line 479: Is there a reference on the soil quality of that area?

Lines 487-489: Please add references